# The Determination of the Limit Load Solutions for the New Pipe-Ring Specimen Using Finite Element Modeling

**Andrej Likeb and Nenad Gubeljak \***

Faculty of Mechanical Engineering, University of Maribor, Smetanova 17, SI-2000 Maribor, Slovenia; andrej.likeb@gmail.com

**\*** Correspondence: nenad.gubeljak@um.si

**Abstract:** To estimate the acceptable size of cracks and predict the loading limit of the pipeline or its resistance to the initiation and crack growth by following the structural integrity, the fracture toughness and limit load solutions are required. Standard fracture toughness testing of thin-walled pipelines is often difficult to perform in order to complete standard requirements. To find an alternative technique for the measurement of the fracture toughness of the already delivered pipeline segment, the new pipe-ring specimen has been proposed; however, the limit load solutions have not been investigated yet. The limit load depends on the geometry of the specimen and loading mode. The ligament yielding of pipe-ring specimens containing axial cracks through the thickness under combined loads was calculated by the finite element method. This paper provides limit load solutions of several different pipe-ring geometries containing two diametric symmetrical cracks with the same depth ratio in a range of $0.45 \leq a/W \leq 0.55$. The limit load (LL) solutions calculated by numerical analysis are shown as a function of the full ring section's size and the corresponding crack aspect ratio for determining the normalized load. These can potentially construct the failure assessment diagram to estimate the crack acceptance in a part of the pipe.

**Keywords:** limit load; pipe-ring specimen (PRS); finite element analysis; thin-walled; loading of pipelines

## 1. Introduction

Pipelines, besides all types of vehicular transport, represent the most common way of transporting natural gas, and hence require a lot of attention mostly for ensuring a reliable and safe supply. Transmission thin-walled pipelines, the most common globally used pipelines, used over very long distances, for which pipes are usually welded together from smaller, 25–30 m-long segments, and dug into the soil, are very difficult to maintain and provide with structural integrity mechanisms to prevent catastrophic situations, such as gas leakage from a small material defect. From the view of ensuring the structural integrity of the pipelines, knowing the mechanical and the fracture mechanics properties of pipeline material is crucial for safe, stable, and reliable usage. The standard way of measuring fracture toughness in the pipelines in accordance with ASTM E-1820 [1] and BS 7440 [2] standards is inadequate because the standard approach requires time-consuming testing and very expensive manufacturing of the standard specimens from the pipe wall. Another reason why the standard way of measuring fracture toughness is not recommended is that during testing, the triaxial stress state with different constraint effects appears. Consequently, the fatigue crack grows non-uniformly throughout the thickness of the standard compact tension (CT) specimen [3,4]. The literature [4,5] shows a detailed study on the suitability of a ring specimen to determine

the fracture toughness of a pipeline material. The authors presented and discussed the results of the research at international and domestic professional conferences [6,7]. Testing the fracture toughness and the material behavior is most important to be performed in the axial (longitudinal) direction of the pipeline, due to twice as high stresses in the radial (perpendicular) direction compared to the axial direction. The reason for such stress allocation is longitudinal pipeline opening when the gas starts to leak, causing an explosion. In the case of micro-crack nucleation, if it becomes critical that causes larger gas leakage and the guaranteed explosion can cause a crack to grow with up to 250 m/s; more for ductile materials and exceeding 600 m/s in the case of brittle materials. This proves just how important it is to know the material properties, loading modes, and the geometry of the analyzed component to estimate the critical fracture toughness parameters and hence to prevent such a situation before it is too late. For an alternative means of measuring the fracture toughness of pipelines a new specimen called the pipe-ring specimen (PRS) has been proposed. A new specimen is a simple ring easily cut from a segment of the gas pipe and loaded statically with the three-point bending test on a hydraulic machine. The only dimension we must follow is a standard ratio of height to wall thickness, which is 2, i.e., the ring's cutout length must be twice the wall thickness [1,2]. The measurement relies on the use of size appropriate clip gauges of Crack Opening Displacement- COD extensometers. The base comparison stands on the idea to replace the testing with a standard Single Edge Notched Bending-SENB specimen, which is three-point bending tested. Although, the best comparison to verify the new specimen would be with the standard CT specimens, except for the problem of geometry; all other stresses and the test conditions are similar to the axial opening of the pipe, so we focused on SENB specimens only. In order to perform the measurement, the PRS specimen must contain a machine-made notch or crack with a crack aspect ratio *a/W* in the range from 0.45 to 0.55, which is the standard recommendation. To accomplish and verify the new approach of testing, many experiments with PRS specimens were done and still remain to be done before we will be able to give complete and final conclusions of the acceptability to use this kind of new specimen. The fracture toughness measurement for the PRS specimen has been already presented and was shown in several previous studies [8] and [9]. The limit load (LL) is the main input parameter for the measurement of the proximity to plastic collapse. Limit loads are analyzed and used in several international EU FP5 up to FP7 projects, such as SINTAP [10] and FITNET [11]. The procedure for determining the limit load is described in [12] and has been used successfully for local and global solutions [13]. Furthermore, the approach proposed by Ainsworth [14] has been successfully applied in BS 7910 [15].

Until now, no analytical or empirical limit load solutions were done for through-wall cracks for specimens such as PRS, especially where it concerns combined bending and shear loads. Limit load solutions based on multi-parametric analysis, as it was parametrically perform in [16]. However, solutions for limit load can be applied in the structural integrity assessment of different components made as pipes or hollow cylinders or components, where a longitudinal crack can occur. An example of the use of limit load solutions on cylindrical and spherical parts is presented in the literature [17,18]. Meanwhile, in the case of solving the integrity of components [19] and welds [20], we need to know the limit load.

Stress strain behavior is simulated by numerical simulations using the finite element method [21,22]. With the help of numerical simulations, the maximum load of elastoplastic pipelines can be determined very precisely [22].

The goal of this paper is to show the approach of determining the LL solutions using finite element modelling (FEM) calculations in the range of different three-dimensional crack depths and different ring dimensions of ring size *R/B* = 5 to 20. Mostly we focused on a thin-walled pipe-ring specimen *R/B* = 20 which, under axial three-point bending, more accurately described the behavior of real thin-walled pipelines. The determination is based on the analytical stress expression for the full section of a wall, with a numerically corresponding function for a different crack depth, in order to investigate the applicability of

the limit load solutions. We named the most critical analyzed section section A, which stands on the plane of load line displacement [4,5].

## 2. Analytical Determination of the Stress Equation for Solid Section of Wall

Solving the stress-strain term of the axially three-point bend ring was necessary for the determination of empirical equations of the limit load for the three-point bend ring in the axial direction. The analytical solution approach was done on the basis of the engineering strength terms for bending, shear, and torsion. As is known, due to the geometry of the ring during loading, all bending, shear, and torsional stresses are present in both wall sections A and B (Figures 1 and 2). However, for analyzing the ring loading capacity, it was important to only analyze section A. Meanwhile, section B, standing on the place of the supports, is subjected to completely torsional loads, as evidenced by numerical simulations, as was already mentioned in [4–8]. The stress Equation (1) for determining the bending and shear stresses at point 1 of section A is schematically shown in Figure 2a where the dominant bending stresses are.

$$\sigma_A = \sqrt{\left(\frac{6 \cdot F \cdot t \cdot e}{B \cdot W^3}\right)^2 + 3 \cdot \left(\frac{F}{2 \cdot B \cdot W}\right)^2} \tag{1}$$

$$t = 0.9 \cdot R \tag{2}$$

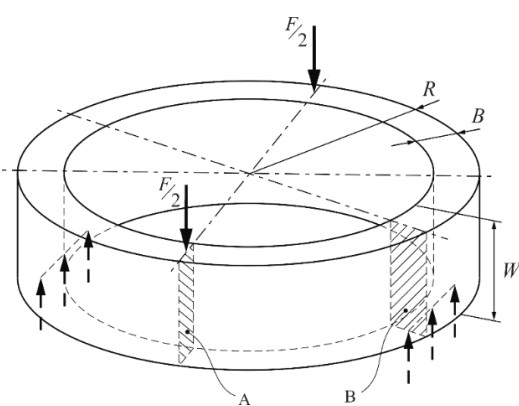

**Figure 1.** The scheme of the ring bending in the axial direction, with both sections A and B [4–8].

In Equation (1), *F* represents the loading force on each half of the ring, and acts as *F*/2, while on the side of supports, it is *F*/4. In other words, the force *F* from Equation (1) is actually $\frac{1}{2}$ of the load on the testing machine where we axially test the ring with three point bending. *t* represents the torque lever of the working force *F* for the displacement from the support, to cross-section A, as seen in Equation (2). The span distance between supports is equal to $2t = 1.8\ R$ (Figure 3). *e* represents the distance to the highest bending stresses with respect to the neutral axis of the section ($B \times W$). Because the stresses in cross section B were not important to design the empirical terms of the limit load for section A, this part is not listed. The previous comparisons with numerical stress estimations in point 1 of section A for several different ring geometries show that the stress Equation (1) is acceptable enough for the calculation and the determination of the limit load of a full section for one sidewall of the ring. The deviation of the linear part of the stress-strain curve up to the yield point is less than or ≈5%, even though the numerically obtained stress was Mises equivalent with included torsional stresses, which are not covered in the analytical term.

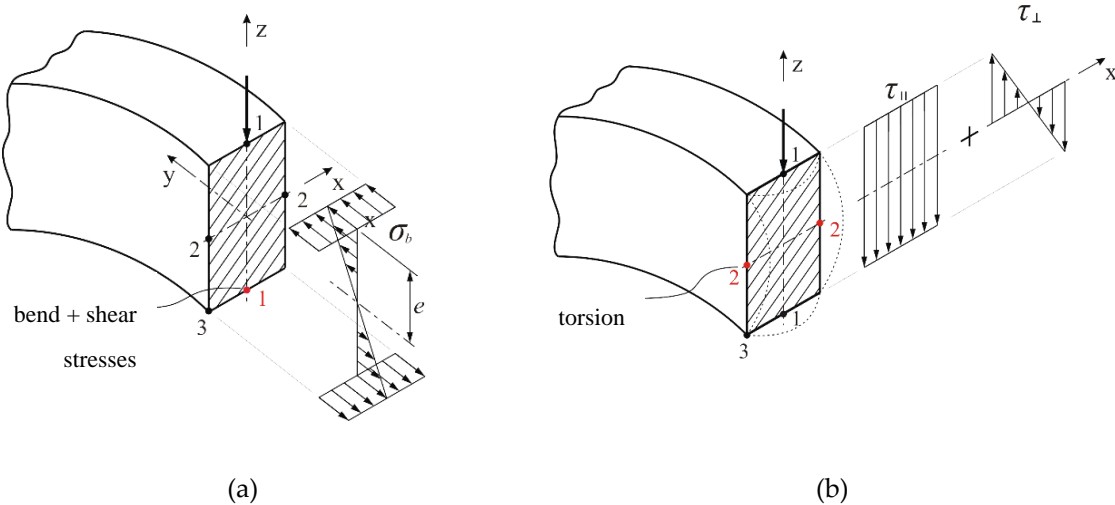

(a)                                                                                      (b)

**Figure 2.** (**a**) Schematic view of the distribution of the bending stresses to the critical cross-section A and (**b**) schematic view of the distribution of both tangential and torsional stresses on the critical cross-section A [4–8].

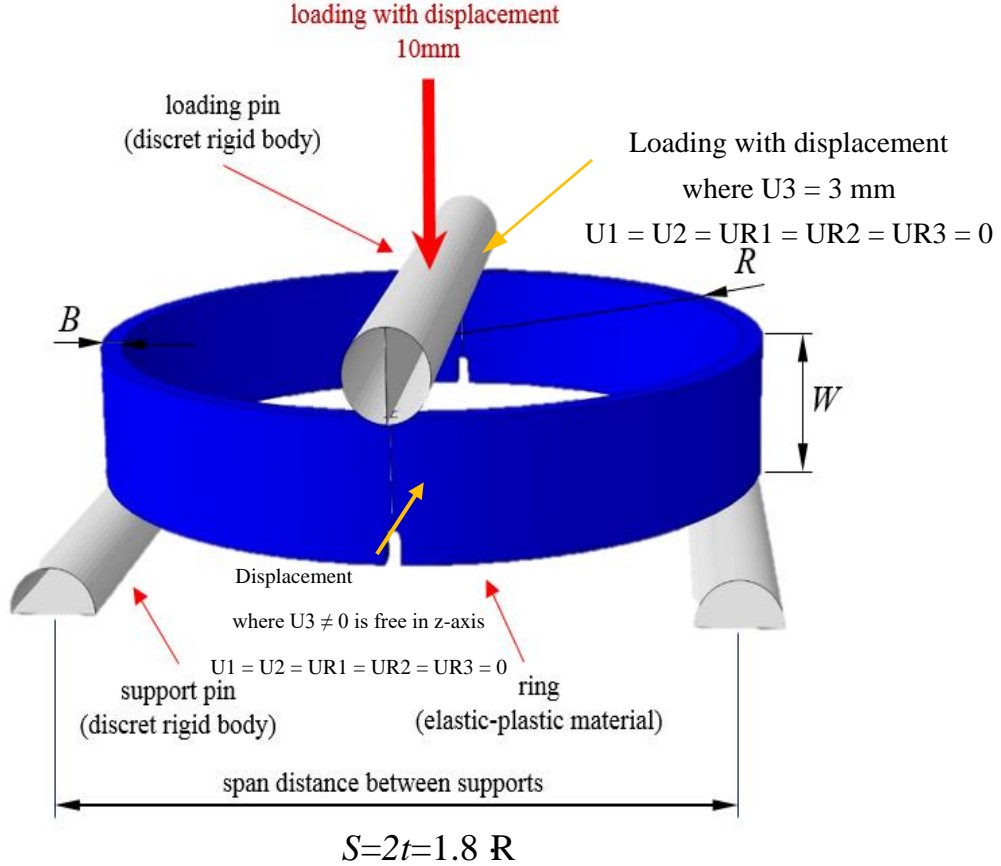

**Figure 3.** Schematically shown geometry of the ring $R/B = 20$, B = 4 mm, W = 8 mm, the span distance between the supports for FEA. The loading mode is three point bending, the scale of the sketch and the dimension do not correspond to the actual ring size).

## 3. Finite Element Elastic-Plastic Analysis

If we want to predict the remaining load limit of the structural components and want to determine the resistance of the material against initiation and crack growth (for the case of damaged pipelines or similar damaged structures), it is necessary to follow Structural Integrity Assessment Procedures for European Industry—SINTAP [10] or FINET [11]—procedures and define the limit load in the dependence of the crack length, to construct the failure assessment diagram (FAD), and predict critical crack length. According to the theory of structure integrity, the collapse occurs when the limit load value is exceeded. The limit load represents the main input parameter for determining plastic collapse [12,13]. The most common way to analyze the marginal state of a structure is using FEM, where, with elastic-plastic analysis of the material, we can directly determine the value of the limit load $F_Y$ [4–8]. FITNET and R6 [14] procedures, as well as BS 7448 and BS 7910 [15] standards, provide a normalized load or a stage of structural plastification with the $L_r$ term (Equation (3)). The $\sigma_{ref}$ is a reference stress of the component with a crack, and $\sigma_Y$ is the yield strength of the material. Normalized load can also be written as the ratio of the current load $F$ on the component and the limit load $F_Y$ of the component.

$$L_r = \frac{\sigma_{ref}}{\sigma_Y} = \frac{F}{F_Y} \tag{3}$$

By the theory, the limit load is known as the load at which the rest of the unbroken ligament is completely plastified. That means at a certain load, the plasticity extends over the unbroken ligament from the deepest point of the crack. Then the limit load appears, and full plastification of the component is the first phase of the component's failure [4–7]. The limit load is dependent on the resistance of the material to plastification and depends on the geometry of the components and configurations of the cracks. It is known that the limit load (indicated by $F_Y$) is lower in the plane stress state than it is in the plane strain state [4,5]. In the case of the plane stress state, plastification appears more distinctly at the height of the limit load compare to the case of the plane strain state, where there is no distinct plastification, and hence an appropriately sized plastic zone of elastic–plastic transition during the loading of the material. The limit load determination can be done by observing the specimen during loading. At a specific time, the LL is marked as the force when the plasticity of the crack is extended over the rest of the unbroken ligament. In the case of the ring size $R/B = 8$, the function of the limit load was performed for the extended crack aspect ratio $a/W$ ranging from 0.3 to 0.8. This function is the starting point for other ring sizes and it represents one point which gives us the margin towards thin-walled pipelines (the ring $R/B = 20$). We have schematically shown a way of analyzing rings as three point bending with outer radius $R$, height of the ring $W$, and wall thickness $B$ in Figure 3. The material properties of the analyzed model were obtained from the tensile tests. The whole research was divided into three main fields: experimental testing, analytical solving, and numerical analysis. Using experimental testing, we characterized the testing material used as a very ductile material with a significant Lüders Plateau at around 470 MPa yield strength [4,5]. Figure 4 shows the actual stress-strain curve with a limit yield of the material $\sigma_{eH}$ = 470 MPa, Young's modulus of elasticity 210 GPa, a Poisson number of steel 0.3, effective fracture stress $\sigma_f$ = 650 MPa at 13% deformation, and a linear approximated actual stress $\sigma_m$= 1842 MPa at 186% deformation. The exponent of hardening of the material is $n$ = 0.4072.

The meshing of the model was done manually by changing the size of the linear quadratic finite elements and their density in the required places for the most accurate results of the cross-section A. The singular allocation of mesh elements around the crack of the model was not performed, because of focusing on just reaching the force at which the material starts to yield. The size of the elements varied from 0.1 to 0.35 mm in the vertical direction and was fixed at 0.3 mm in the horizontal direction. The linear hexagonal elements in the element library of ABAQUS 6.11-3 were used in this analysis. The element

size was 0.15 at the crack front and gradually increased to 0.5 from the crack front towards the edges, using so-called continually increasing element size, as is shown in Figure 5. The shape of the elements was linear hexagonal, because this type of element is appropriate and does not consume more computation time. Other edges of the model were meshed with the number of elements at 100 and a bias ratio of 75. It was modelled with more than 2.3 million of elements, at quarter model. Figure 3 shows the boundary conditions on the lower (fixed) support, too. The properties of the interaction between the loading pin and the specimen, as well as between the support pin and the specimen were required to be introduced in the model. The discretization method was based on surface to surface contact with no adjustments for surfaces. Contact properties were described by a normal component with disallowed separation and by a tangential component with a friction coefficient of 0.1. As mentioned, a numerical analysis was carried out using a dynamic implicit procedure over a time step with period 1. The increment was set to automatic, and the simulation was performed with a maximum number of increments set to 1000. The initial and maximum increment sizes were taken as 0.01. In our study, we manually made a mesh with continuing remeshing of finite element size. We checked that stress goes smoothly in the whole volume of specimens. We have made few variations of elements, but similarly to ref. [23], the sensitivity analysis shows the extremely negligible influence of remeshing when the mesh is established in the proposed way.

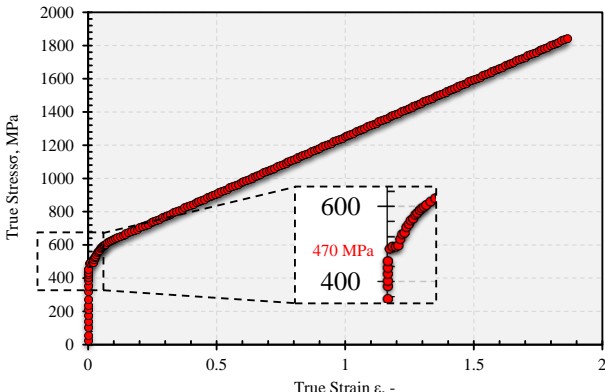

**Figure 4.** True stress-strain curve as the material input properties for determine the numerical analysis.

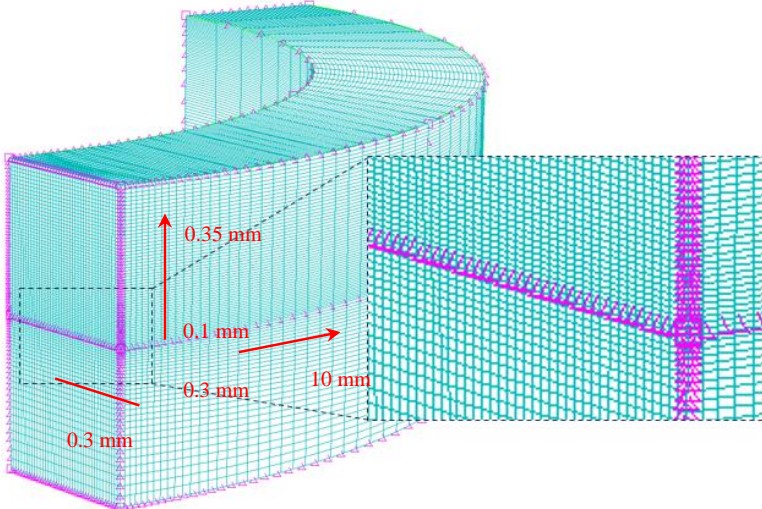

**Figure 5.** The finite elements mesh on $\frac{1}{4}$ of the model, with the displayed places of densification with structural hexagonal linear elements.

## 4. The Ring (*R*/*B* = 20) (Thin-Walled)

According to several references [15], the margin to start calling the pipelines thin-walled pipelines is at a size ratio beyond *R*/*B* = 20. This particular ring shows the thin wall pipelines with ratios of 30 or 40 and more. By analyzing the ring and checking each stage of loading, the first finding was that in the case of the full sectioned ring, the deformation at a displacement of 5.7 mm and corresponding force 1.8 kN starts to spread along the edges (upper and lower) (Figure 6). In the inner layer, the neutral zone of bending stays undeformed, which means the deformation has not spread inside at all, until the more significant effect of local deformation because of the loading pin and push causes the deformation to start spreading inside at 8.5 mm up to a final displacement of 10 mm and a load of 2 kN, exceeding the limit load. This complete deformation is shown by Figure 7, where a very small layer of the neutral zone bending can be seen. The layer which is not fully deformed and also indicates the lower part of the ring starts to bend and rotate inside, because the greater uncompressed layers are located on the inner instead of on the outer side. The fulfilled criterion of plastification is shown in Figure 8a for a crack depth *a*/*W* = 0.5 (the deformation from the crack tip is joined with the deformation from the contact site). The criterion is met already at 0.603 kN, merely 21.5% of the maximum load. Meanwhile, in Figure 8b, this criterion is not met on the outer side of wall.

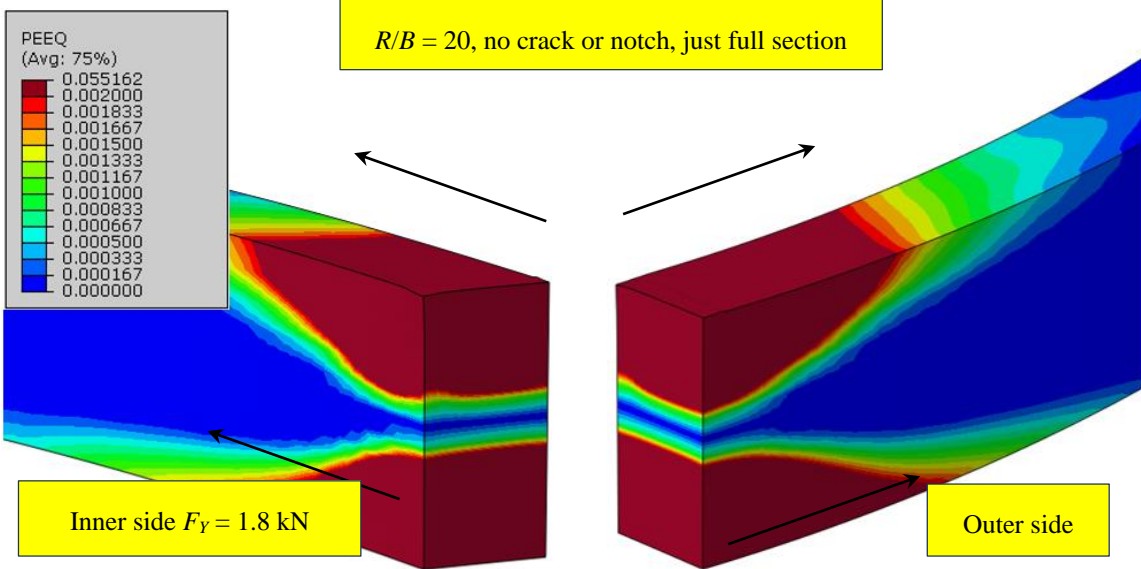

**Figure 6.** At a full cross-section ratio *R*/*B* = 20, the equivalent deformation starts to expand on the upper and lower edge at 5.7 mm displacement, instead of into the middle. The middle neutral layer remains undeformed.

It is known that due to the geometry of the ring, a spatial bending occurs (normal bending and bending on the inner side). Tangential and torsional loads are minimal, and therefore negligible in the cross-section A. This process is hidden in the spatial bending of the ring, because in addition to the normal bending of the ring wall, buckling of the wall to the inner side is also present. The bending to the inner side results in higher stresses on the inner than the outer side. Therefore, the criterion for the limit load is met sooner on the inner side compared to the outer side of the ring. The result is a different strain state [7,8]. In all the analyzed cases of a crack aspect ratio from 0.45 to 0.55, the criterion of the limit load has been met at less than $^1/_3$ of the maximum load (corresponding to 10 mm). Meanwhile, in the cases of a full cross-section, *a*/*W* = 0, where the deformation starts to spread to the edges of the ring instead of in the

middle side, the limit load criterion has been met within less than $^2/_3$ of maximum loading. Specifically, for the derivation of the limit load function, we extended the crack aspect ratio from 0.3 to 0.8 for the analysis of the ring $R/B = 8$, just to have a bigger range of crack depths and finally a more accurate solution. The determination of the limit load as well the derivation of functions has been done for each geometry of the ring. The rings were named by size as follows: $R/B = 5$; 6; 7; 8; 9; 10; 12; 15; and the most important for analysis, 20. The criterion we used at the time was that the limit load is detected as the first load occurred at the end of the linear elastic behavior of the material (i.e., the last point before the start of localized plastic bands-the Lüders slip) (Figures 9 and 10) [4,5,7]. The point used for limit load criterion was labeled as the Lüders slip "I".

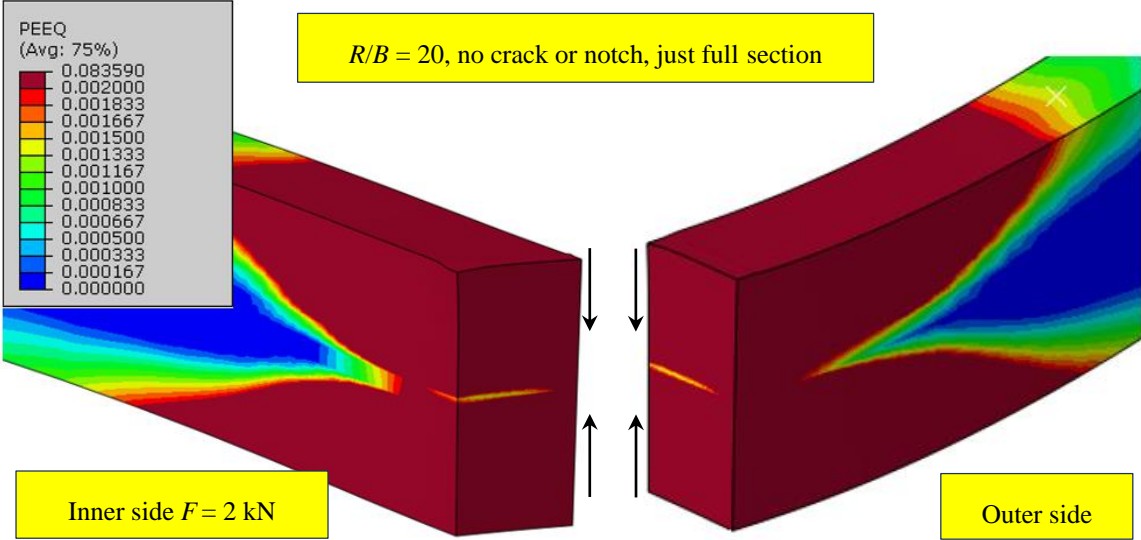

**Figure 7.** The equivalent deformation, where a small layer of the neutral zone still remains undeformed at 10 mm displacement. While spreading of the deformation over the upper and lower periphery of ring stopped, it now started extending into the interior.

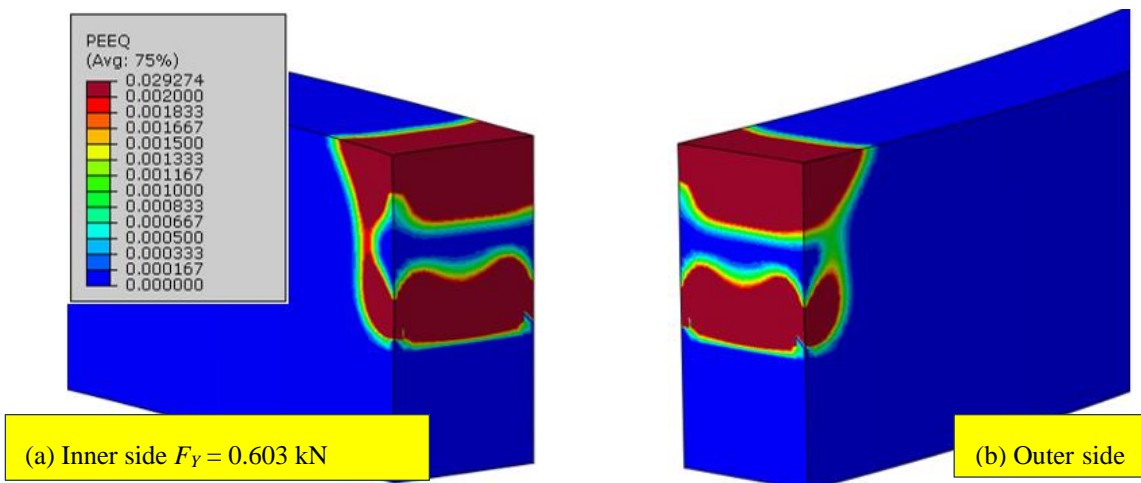

**Figure 8.** The equivalent deformation of (**a**) on the inner side with meeting the criterion of the limit load and (**b**) on the outer side, where the theoretical limit load criterion is not yet met for the ratio $R/B = 20$ at 2.15 mm.

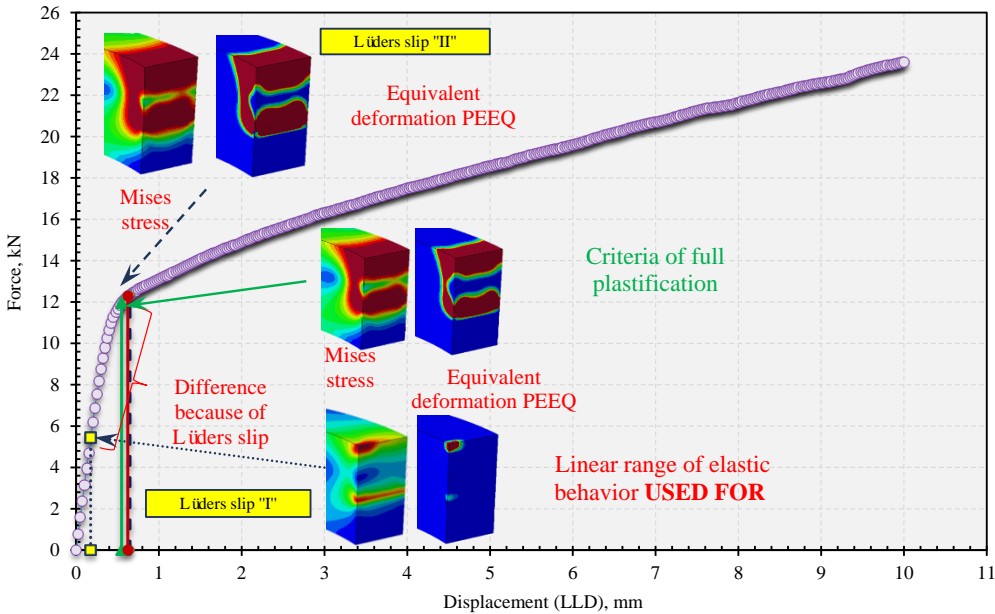

**Figure 9.** The load-displacement curve for a ring with the ratio *R/B* = 5 and a crack depth of *a/W* = 0.45. The figure shows the position of the points where we met specific criteria for determining the limit load on the inner and outer side and for the case of taking the Lüders slip into account.

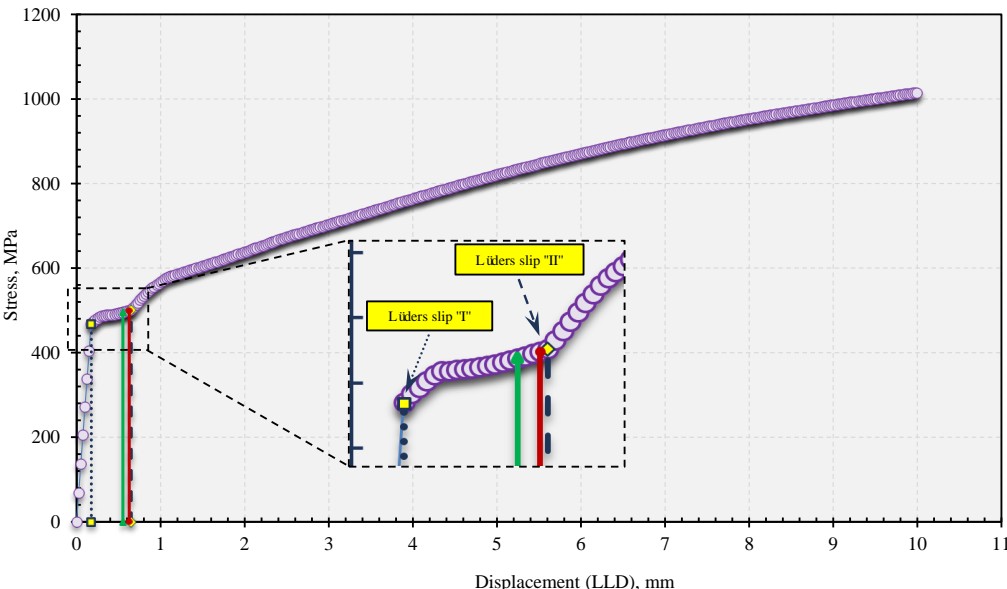

**Figure 10.** Displayed range with points for determining the limit load from the stress-displacement curve (*R/B* = 5 with *a/W* = 0.45) the position of Lüders slip "I" and Lüders slip "II".

The limit load determination is based on the assessment of the position of this point, see Figure 10 (stress-displacement curve of the ring *R/B* = 5, *a/W* = 0.45), and the determination of the corresponding displacement, which we estimated from Figure 9 as $F_Y$. In all cases of the rings we analyzed, $F_Y$ was much lower than from the criterion of full plastification through the unbroken ligament. The deformation at this time was only noted and visible at the contact point (s) [4,5].

## 5. Determination of the Limit Load Function

The determination of the empirical equation and final function of the limit load in dependence of crack length based on the solid (full) ring section (i.e., no cracks or notches) in Equation (1). The derivation for the limit load at point "1" of section A stands by simply equating $\sigma_A = \sigma_Y$ and load $F = F_{Y0}$:

$$\sigma_Y = \sqrt{\left(\frac{6 \cdot F_{Y_0} \cdot t \cdot e}{B \cdot W^3}\right)^2 + 3 \cdot \left(\frac{F_{Y_0}}{2 \cdot B \cdot W}\right)^2} \tag{4}$$

By forward derivation, we get:

$$F_{Y0} = \frac{2 \cdot \sigma_Y \cdot B \cdot W}{\sqrt{3} \cdot \sqrt{1 + 48 \cdot \left(\frac{t \cdot e}{W^2}\right)^2}} \tag{5}$$

Equation (5) is applied for the determination of the limit load on one side of the wall of a solid section of the ring. The limit load of the entire ring with the solid section is equal to $2F_{Y0}$. If instead of moments of inertia we use the resistance moment, then the formula in section A is:

$$F_Y\left(\frac{a}{W}\right) = 2 \cdot \frac{2\sigma_Y \cdot B \cdot W}{\sqrt{3} \cdot \sqrt{1 + 12 \cdot \left(\frac{t}{W}\right)^2}} \cdot \left(f_0 \cdot \left(\frac{a}{W}\right)^0 + f_1 \cdot \left(\frac{a}{W}\right)^1 + f_2 \cdot \left(\frac{a}{W}\right)^2 + f_3 \cdot \left(\frac{a}{W}\right)^3\right) \tag{6}$$

The function of limit load $F_Y$ (a/W) for a pipe-ring included crack or notch is given by Equation (6) as a function of the crack aspect ratio $f$ (a/W). Here $f_0, f_1, f_2$ and $f_3$ represent the coefficients of the third order polynomial trend curve [4–7]. Appendix A displays the equation of the limit load $F_Y$ as well as the normalized limit load after Equation (3), if we have been given the yield strength of a material for an axially three-point bend pipe-ring specimen, by following sizes in Appendix B. The LL function is given depending on the polynomial coefficients created based on the dimensions of the analyzed rings. Figure 11 shows the limit load function of the ring $R/B = 8$ and its standard height vs. thickness $W/B = 2$ depending on the extended crack aspect ratio a/W from 0.3 to 0.8. Meanwhile, all other rings were analyzed for a range from a/W = 0.45 to 0.55.

The comparison of the LL function is for a ratio of 8, with all other ratios from 5 to 20 are displayed in Figure 12. It is obvious that the curves in accordance with the size of ring, do not sequentially follow the ratio $R/B$, because while we were planning the geometry of the rings, we completely randomly defined the outer radius $R$, depending on the ratio $R/B$. It shows that in some analyzed cases, the wall thickness had a much higher bending stress component compared to the torque lever than in other cases. These results show completely different behavior of the ring during loading, since we have to perform analysis with different stress-strain states not just with geometries of the ring, but also by comparing the inner and outer side of the bent ring. Figure 13 shows the distribution of the limit load $F_Y$ for a ring with a crack or notch in relation with the limit load of the solid Section $2F_{Y0}$ depending on the ratio $R/B$ (i.e., the size of the ring to the wall thickness $B$). Figure 13 indicates a very discontinuous ratio $F_Y/2F_{Y0}$ for all three crack aspect ratios in the range of $R/B$ from 6 to 9. Until other investigations are done, such as investigating the constraint effect or analyzing the stress triaxiality, it is not possible to make any comments on the validity of the obtained results in the range of small rings. The described state indicates an uneven stress-strain state throughout the wall thickness as a result of random choice of ring dimensions and corresponding triaxiality. In such behavior, we have to deal also with a bigger effect of the plane strain state, since the thickness is larger compared to the moment arm (torque). The positive side in Figure 13 shows that for the bigger ring size,

from 12 to 20, we have very nice formed continuous curves. This show the suitability for characterizing thin-walled pipelines and analyzing the sizes of rings which comply with natural gas pipelines.

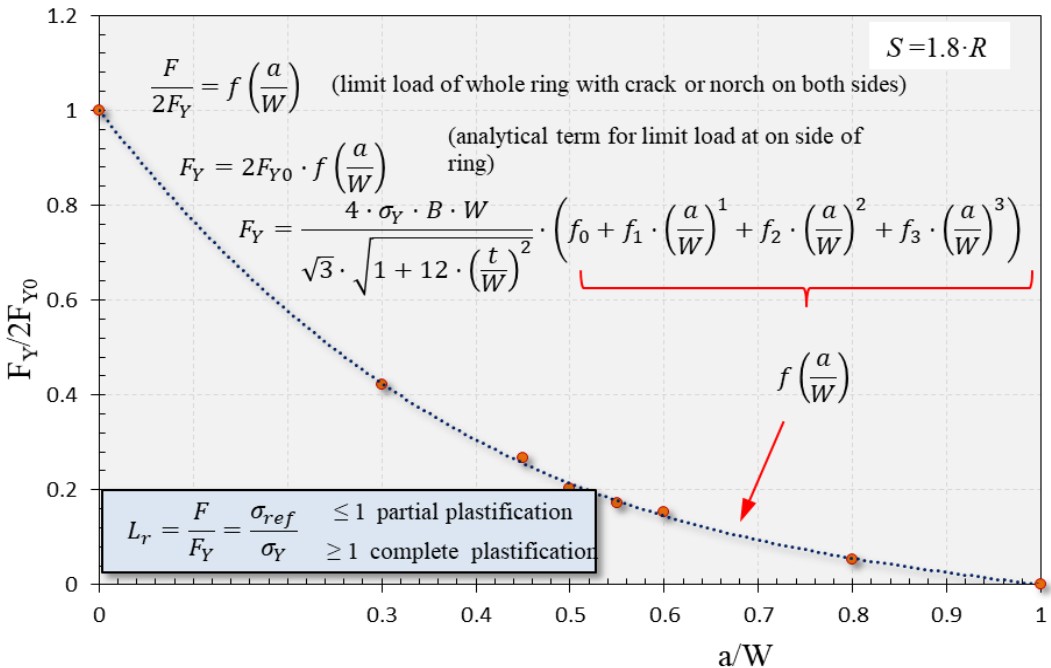

**Figure 11.** The limit load function for $R/B = 8$ and $W/B = 2$ in dependence on different crack aspect ratios $a/W$ from 0.3 to 0.8.

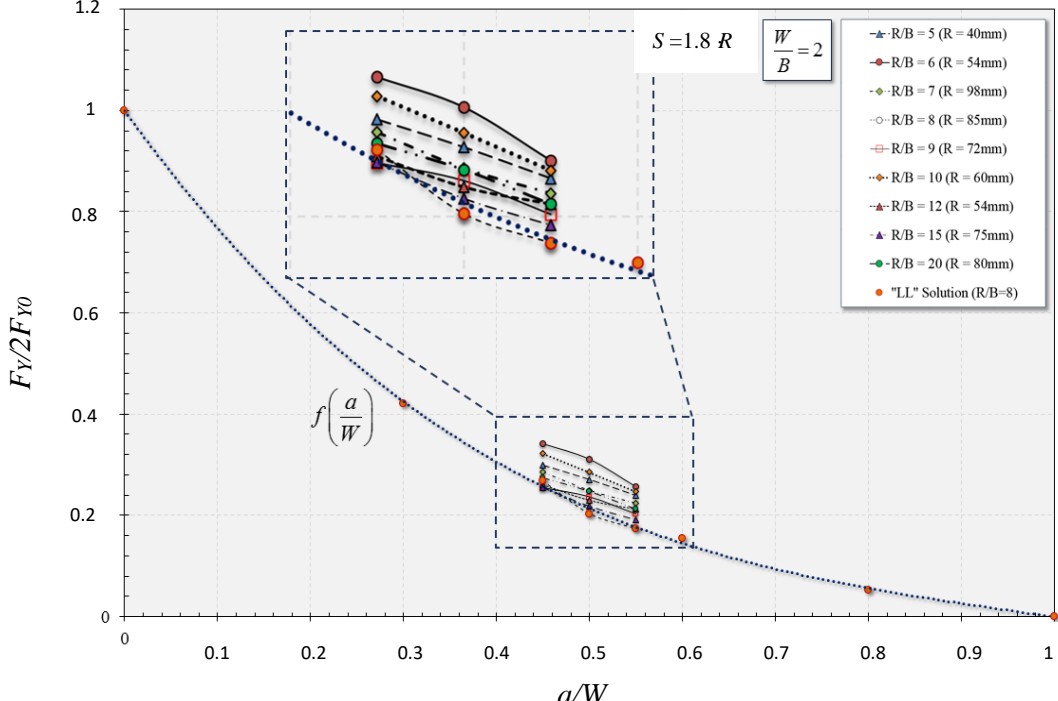

**Figure 12.** Comparison of matching the limit load function of $R/B = 8$ with other ring sizes in the range of the standard crack aspect ratio $a/W = 0.45$ to 0.55 [4–7].

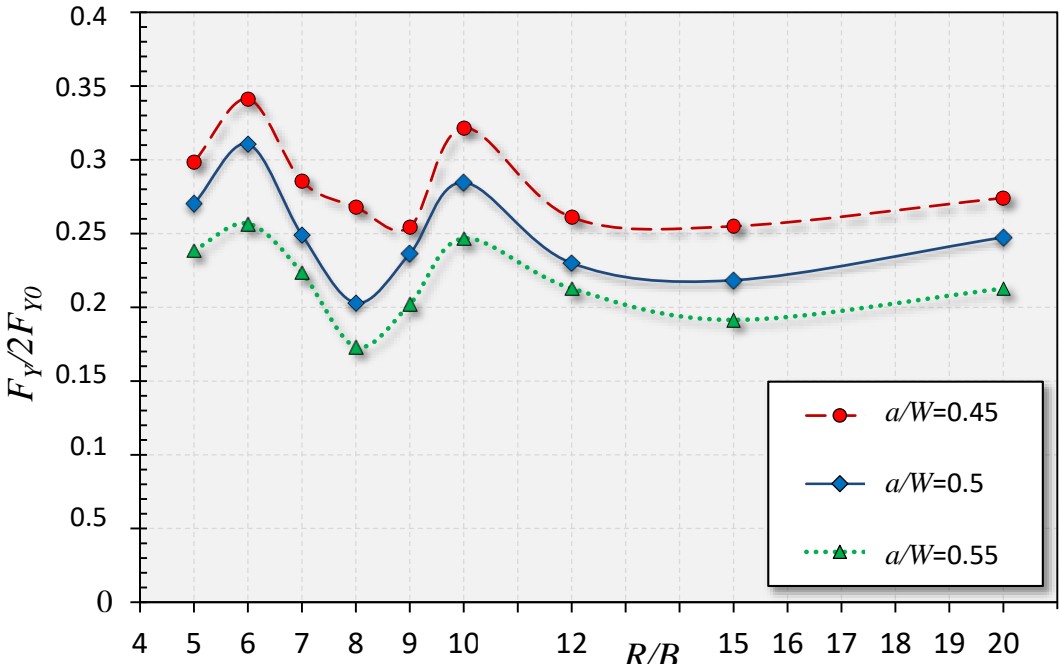

**Figure 13.** Obtained values of the limit load for all geometries from 5 to 20 and a crack or notch depth *a/W* in relation to the limit load of the same geometry in a solid section ring [4–7].

## 6. Conclusions

As reiterated in this article and well known for dynamically exposed structures, the limit load (LL) in addition to the stress intensity factor (SIF) are the most important quantities for structural integrity to predict the maximum load capacity and possibly the largest "allowed" crack size for material safety in regard to other exposure conditions. In the frame of the focused research here, we were mostly verifying that the new proposed ring specimens, the so-called pipe ring notch bend specimens (PRNB), can be used for measurement of fracture toughness. In one of the many required parts of this research, we calculated the ligament yielding point on the stress-strain curve using FEA with the ABAQUS non damage model for different ring sizes. This paper shows the procedure we chose to define and create an engineering practice to calculate the limit load of the axial three-point bend ring, which is simply cut from the segment or just a small piece of pipeline.

- The limit load solutions of randomly chosen several different geometries, show interesting non continuous behavior depending the on the ring size, i.e., the ratio of its diameter and wall thickness, *R/B*, from 6 to 10.
- A plausible reason is the randomly picked diameter of the corresponding ring size ratio *R/B*, while in the case of more reasonable ring sizes with an increasing diameter, including dimensions taken from real pipelines, with correspondingly increasing thickness, we can expect that the LL values would slightly rise with the increasing of the ring's size ratio *R/B*, for all three crack or notch depth *a/W* cases, as seen in Figure 13 for *R/B* from 12 to 20 and above, to the ratio of real-world thin-walled pipelines.
- The function we expressed to define LL in a range of different ring sizes is calculated for various crack aspect ratios from 0.45 to 0.55. We also calculated the extended range of the crack aspect ratio from 0.3 to 0.8 for one randomly chosen probe ratio *R/B* = 8, where the span distance between the supports is 1.8 times *R*, just to schematically show the calculation of the limit load if the notch or crack is not in the range of standard recommendations.

- By observing and processing all numerical results, we found spatial bending of all probes subject to the different constraint effects of limiting and spreading the yield deformation around the tip, and along the crack path. However, as we noted, stress triaxiality needs to be analyzed for a better footing to explain and completely describe the behavior of axial three-point bend ring probes.

**Author Contributions:** Conceptualization, A.L.; Investigation, A.L.; Methodology, N.G.; Project administration, N.G.; Software, A.L.; Supervision, N.G. All authors have read and agree to the published version of the manuscript.

**Funding:** This research received no external funding.

**Acknowledgments:** Authors acknowledge to Slovenian Research Agency (ARRS) for financial support of Ph.D. investigation of Andrej Likeb and Reseach Program P2-0137 "Numerical and experimental analysis of mechanical systems".

**Conflicts of Interest:** The authors declare no conflict of interest.

## Appendix A

The general equation for calculating the limit load by the corresponding coefficients for the ratio of the size of the ring.

| The general equation for calculating the limit load in dependence of the crack depth $a/W$ from 0 to 1 and ratio of ring's size of $R/B = 8$ and other ratios $R/B$ from the $a/W = 0.45$ to 0.55. |
|---|
| $$F_Y\left(\tfrac{a}{W}, \tfrac{R}{B}\right) = 2 \cdot \frac{2\sigma_Y \cdot B \cdot W}{\sqrt{3} \cdot \sqrt{1 + 12 \cdot \left(\tfrac{t}{W}\right)^2}} \cdot \left(f_0 \cdot \left(\tfrac{a}{W}\right)^0 + f_1 \cdot \left(\tfrac{a}{W}\right)^1 + f_2 \cdot \left(\tfrac{a}{W}\right)^2 + f_3 \cdot \left(\tfrac{a}{W}\right)^3\right)$$ |

Nomenclature:

| | |
|---|---|
| $F_Y$ | - limit load of whole ring, N |
| $a$ | - crack length (depth), mm |
| $W$ | - height of the ring cut from the pipe, mm |
| $R$ | - outer radius of the ring, mm |
| $B$ | - wall thickness of the ring, mm |
| $t$ | - the moment arm, mm, Equation: $t = 0.9R$ (Figure 3) |
| $\sigma_Y$ | - the yield strength (proportional) of the material, MPa |
| $f_0, f_1, f_2, f_3$ | - polynomial coefficients |

| R/B | $f_0$ | $f_1$ | $f_2$ | $f_3$ |
|---|---|---|---|---|
| 5 | 0.3988 | 0.0835 | −0.6812 | - |
| 6 | −0.4321 | 3.8168 | −4.6626 | - |
| 7 | 1.0998 | −2.7835 | 2.1638 | - |
| 8 | 0.9995 | −2.5552 | 2.3803 | −0.8256 |
| 9 | −0.303 | 2.6802 | −3.2033 | - |
| 10 | 0.6145 | −0.5682 | −0.1831 | - |
| 12 | 1.1626 | −3.2481 | 2.7656 | - |
| 15 | 1.0233 | −2.585 | 1.9502 | - |
| 20 | 0.1511 | 1.0001 | −1.6148 | - |

## Appendix B

**Table A1.** The geometry configurations of the numerically modelled rings in the dependence by the ratio R/B between the outer radius and the wall thickness and crack depth *a/W*.

| R/B | a/W | R, mm | W, mm | B, mm | a, mm |
|---|---|---|---|---|---|
| **5** | - | 40 | 16 | 8 | - |
| | 0.45 | 40 | 16 | 8 | 7.2 |
| | 0.5 | 40 | 16 | 8 | 8 |
| | 0.55 | 40 | 16 | 8 | 8.8 |
| **6** | - | 54 | 18 | 9 | - |
| | 0.45 | 54 | 18 | 9 | 8.1 |
| | 0.5 | 54 | 18 | 9 | 9 |
| | 0.55 | 54 | 18 | 9 | 9.9 |
| **7** | - | 98 | 28 | 14 | - |
| | 0.45 | 98 | 28 | 14 | 12.6 |
| | 0.5 | 98 | 28 | 14 | 14 |
| | 0.55 | 98 | 28 | 14 | 15.4 |
| **8.5** | - | 85 | 20 | 10 | - |
| | 0.45 | 85 | 20 | 10 | 9 |
| | 0.5 | 85 | 20 | 10 | 10 |
| | 0.55 | 85 | 20 | 10 | 11 |
| **9** | - | 72 | 16 | 8 | - |
| | 0.45 | 72 | 16 | 8 | 7.2 |
| | 0.5 | 72 | 16 | 8 | 8 |
| | 0.55 | 72 | 16 | 8 | 8.8 |
| **10** | - | 60 | 12 | 6 | - |
| | 0.45 | 60 | 12 | 6 | 5.4 |
| | 0.5 | 60 | 12 | 6 | 6 |
| | 0.55 | 60 | 12 | 6 | 6.6 |
| **12** | - | 54 | 9 | 4.5 | - |
| | 0.45 | 54 | 9 | 4.5 | 4.05 |
| | 0.5 | 54 | 9 | 4.5 | 4.5 |
| | 0.55 | 54 | 9 | 4.5 | 4.95 |
| **15** | - | 75 | 10 | 5 | - |
| | 0.45 | 75 | 10 | 5 | 4.5 |
| | 0.5 | 75 | 10 | 5 | 5 |
| | 0.55 | 75 | 10 | 5 | 5.5 |
| **20** | - | 80 | 8 | 4 | - |
| | 0.45 | 80 | 8 | 4 | 3.6 |
| | 0.5 | 80 | 8 | 4 | 4 |
| | 0.55 | 80 | 8 | 4 | 4.4 |

Meaning of the labels: *R/B*—size ratio of the ring, *R*—outer radius of the ring, *W*—height of the ring, *B*—thickness of wall, *a*—crack or notch length.

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
