# Peer review of "The Determination of the Limit Load Solutions for the New Pipe-Ring Specimen Using Finite Element Modeling"

_metals, doi:10.3390/met10060749_

Round 1

Reviewer 1 Report

The publication was written correctly. The studies presented in the paper are current scientific problems. The paper present determination process of the limit load for new Pipe-Ring specimen using FEM. In my opinion, the work is suitable for publication if certain comments and suggestions are included.

  1. In the introduction section, the cited literature should be presented in order from 1 to 20. Already on page 1 of the publication it is visible that in 34 lines of the text, the literature position 1 and 2 was cited, while in 39 lines of the text, the literature position 4 and 5 was cited. It is necessary to take care of the order and rebuild the literature in such a way that the numbering is in turn - and item 3 is missing.
  2. In the 63 and 68 lines of text in the introduction, citations should be divided into a higher number of descriptive sentences - not one sentence citing several literary items.
  3. The designations of formulas 1 and 2 on page 3 should be aligned to the right. The same applies to the formulas on pages 4 and 10 - with different fonts in the description of the formulas on page 10, which is not correct.
  4. Figure 3 is presented in the wrong way, the markings are not legible, the descriptions are not correctly positioned - please make the necessary corrections and make sure that this figure is legible.
  5. On Figures 4a and 4b graphical errors are visible - on the horizontal axis of Fig. 4a the description of the digit 0.13 should be corrected, while on the approximate area of Fig. 4b there is some red text.
  6. There is no information in Chapter 3 about the number of finite elements and computational nodes, thus presenting the size of the computational issue.
  7. On page 11 in Figure 11 the description of the horizontal axis is illegible because it contains some graphical errors, please take care of better readability. The same applies to Figure 12 only that in the description of the legend. Please refer to the similar notes on page 12 in Figure 13.
  8. The text on page 12 is written in two different font sizes, please standardize it.

After having commented on all of the above, I think that the publication will be strengthened accordingly and may be published in the journal Metals.

Author Response

Dear reviewer,

Thank you for your reviewing and comments. We have taken your opinion into account and corrected it in the text as follows:

Points 1 and 2.

We have now corrected this and renumbered the references.

Literature [4,5] shows a detailed study on the suitability of a ring specimen to determine the fracture toughness of a pipeline material. The authors presented and discussed the results of the research at international and domestic professional conferences [6,7].

Limit loads are analyzed and used in several international EU FTP 5 up to FTP7 projects, such as SINTAP [10] and FITNET [11]. The procedure for determining the limit load is described in [12] and has been used successfully for local and global solutions [13]. Furthermore, the approach proposed by Ainsworth [14] has been successfully applied in BS 7910 [15].

An example of the use of limit load solutions on cylindrical and spherical parts is presented in literature [17 and 18]. Also in the case of solving the integrity of components [19] and welds [20], we need to know the limit load.

  1. We have rewriten formulas with the same fonts and designated formulas with right aligned numbers.
  2. Thank you. We have rearanged the markers of Fig.3.
  3. We have removed Fig. 4.a) and only kept the plot of Fig.4.b) as Figure 4, because in the text we have explained the true stress-strain diagram.
  4. We have taken your opinion into account and corrected the text as follows:

The meshing of the model has been done manually by changing the size of the linear quadratic finite elements and their density in the required places for the most accurate results of the cross-section A. The singular allocation of mesh elements around the crack of the model was not performed, because of focusing on just reaching the force at which the material starts to yield. The size of the elements varied from 0.1mm to 0.35mm in the vertical direction and was fixed at 0.3mm in the horizontal direction. The linear hexagonal elements in the element library of ABAQUS 6.11-3 were used in this analysis. Element size was 0.15 at the crack front and gradually increased to 0.5 from crack front towards the edges, using so-called continually increasing element size. The shape of the elements was linear hexagonal, because this type of elements is appropriate and does not consume more computation time. Other edges of the model were meshed with the number of elements at 100 and a bias ratio of 75. It was modelled with 2.345.236 elements. Figure 3 shows the boundary conditions on the lower (fixed) support. The properties of the interaction between the loading pin and the specimen, as well as between the support pin and the specimen were required to be introduced in the model. The discretization method was based on surface to surface contact with no adjustments for surfaces. Contact properties were described by a normal component with disallowed separation and by a tangential component with a friction coefficient of 0.1. As mentioned, a numerical analysis was carried out using a dynamic implicit procedure over a time step with period 1. Increment was set to automatic, and the simulation was performed with a maximum number of increments set to 1000. The initial and maximum increment sizes were taken as 0.01.

  1. We have corrected Figs. 11, 12 and 13 according to your suggestions. Thank you.
  2. We have standardized the font size in text on page 12.

Author Response

Dear reviewer,

Thank you very much for your comments and suggestion. According our answers are follows:

  1. We have extended our introduction with recommended literature:

Stress strain behavior is simulated by numerical simulations using the finite element method [21, 22]. With the help of numerical simulations, the maximum load of elastoplastic pipelines can be determined very precisely [22].

  1. Are you referring to Fig. 4.a)? We removed figure 4.a) and present only Fig.4.b) as Fig. 4.

  1. Sensitivity analysis has been performed and we are referring to the suggested reference on follow way:

In our study we have manually made a mesh with continuing remeshing of finite element size. We have check that stress goes smo0thly in whole volume of specimens. We have made few variations of elements but similarly as in ref. [23], the sensitivity analysis shows the extremely negligible influence of remeshing when the mesh is established in the proposed way.

  1. We explain about the mesh type in our model in the following way:

The meshing of the model has been done manually by changing the size of the linear quadratic finite elements and their density in the required places for the most accurate results of the cross-section A. The singular allocation of mesh elements around the crack of the model was not performed, because of focusing on just reaching the force at which the material starts to yield. The size of the elements varied from 0.1 mm to 0.35 mm in the vertical direction and was fixed at 0.3 mm in the horizontal direction. The linear hexagonal elements in the element library of ABAQUS 6.11-3 were used in this analysis. Element size was 0.15 at the crack front and gradually increased to 0.5 from crack front towards the edges, using so-called continually increasing element size. The shape of the elements was linear hexagonal, because this type of elements is appropriate and does not consume more computation time. Other edges of the model were meshed with the number of elements at 100 and a bias ratio of 75. It was modelled with 2.345.236 elements. Figure 3 shows the boundary conditions on the lower (fixed) support. The properties of the interaction between the loading pin and the specimen, as well as between the support pin and the specimen were required to be introduced in the model. The discretization method was based on surface to surface contact with no adjustments for surfaces. Contact properties were described by a normal component with disallowed separation and by a tangential component with a friction coefficient of 0.1. As mentioned, a numerical analysis was carried out using a dynamic implicit procedure over a time step with period 1. Increment was set to automatic, and the simulation was performed with a maximum number of increments set to 1000. The initial and maximum increment sizes were taken as 0.01.

  1. The boundary conditons are now described more clearly and shown in Fig. 3.
  2. We have addresssed parts of the conclusion with bullet points.

Round 2

Reviewer 2 Report

The authors considered the reviewers' comments carefully. Therefore, it is recommended to be published now.